# Overall Survival and Prognostic Factors in De Novo Metastatic Human Epidermal Growth Factor Receptor (HER)-2-Positive Breast Cancer: A National Cancer Database Analysis

**DOI:** 10.3390/cancers17111823

**Published:** 2025-05-30

**Authors:** Meghana Kesireddy, Durva Masih, Valerie K. Shostrom, Amulya Yellala, Samia Asif, Jairam Krishnamurthy

**Affiliations:** 1Division of Hematology and Medical Oncology, Department of Internal Medicine, University of Nebraska Medical Center, Omaha, NE 68198, USAjairam.krishnamurthy@unmc.edu (J.K.); 2Department of Biostatistics, University of Nebraska Medical Center, Omaha, NE 68198, USA

**Keywords:** metastatic HER2-positive breast cancer, overall survival, prognostic factors, national cancer database, demographic factors, clinicopathological factors, treatment factors, Cox proportional hazard model, multivariate analysis

## Abstract

Metastatic HER2-positive breast cancer now has an average five-year survival rate due to new treatments developed over the last decade. Survival varies greatly among patients. To understand these differences, we studied a large group of women newly diagnosed with metastatic HER2-positive breast cancer in the National Cancer Database. Key factors influencing survival include age, comorbidity score, histology, HER2 IHC expression, hormone receptor status, the number and location of metastatic sites, and the types of treatment received, such as first-line chemotherapy or anti-HER2 therapy. Our findings can help predict survival based on individual factors and guide treatment strategies.

## 1. Introduction

Breast cancer ranks as the most commonly diagnosed cancer among women globally, with more than 2 million new cases reported in 2020. In the United States, it represents 29% of all newly diagnosed cancers in women [1]. Breast cancer exhibits significant biological diversity, and patient outcomes are influenced by both the stage at diagnosis and tumor subtype. Clinically, breast cancer is primarily categorized into three main subtypes based on receptor expression: hormone receptor (HR)-positive and HER2-negative, which makes up about 70% of cases; HER2-positive, with or without HR expression, accounting for 15–20% of breast cancers; and triple-negative breast cancer, defined by the absence of both HR and HER2 expression, representing 10–15% of breast cancers [2]. Approximately 15–20% of breast cancers over-express the human epidermal growth factor receptor 2 (HER2) and are known as HER2-positive breast cancers [3]. HER2 is an epidermal growth factor receptor (EGFR) related to tyrosine kinase and is encoded by ERBB2 (erythroblastic oncogene B). HER2 expression is evaluated through immunohistochemistry (IHC) or gene expression analysis via fluorescent in situ hybridization (FISH). A positive HER2 result consists of an IHC score of 3+ or 2+ alongside a positive FISH [4].

Approximately 15% to 24% of patients with localized HER2-positive breast cancer will develop metastatic disease after completing curative-intent therapy for the localized setting, and 3% to 10% present with de novo metastasis [5]. Systemic therapy for metastatic breast cancer aims to extend survival, relieve symptoms, and preserve quality of life. Median survival differs based on tumor subtype, metastatic sites, and overall cancer burden. Advancements in anti-HER2 therapies, beginning with trastuzumab and moving beyond with agents like Trastuzumab emtansine, Trastuzumab deruxtecan, tucatinib, Margetuximab, neratinib, etc., have significantly improved outcomes for HER2-positive breast cancer patients [6]. Despite these treatment advances, many patients ultimately face cancer progression due to the development of treatment resistance.

Currently, the median overall survival (OS) for metastatic HER2-positive breast cancer is nearly five years. Recent results from the CLEOPATRA trial demonstrated a median OS of 57 months, establishing the combination of trastuzumab and pertuzumab with chemotherapy as the current standard first-line treatment for ERBB2-positive metastatic breast cancer [7]. Notably, the CLEOPATRA trial reveals remarkable durable disease control in some patients; 16% of those receiving docetaxel, trastuzumab, and pertuzumab are alive and progression-free after 8 years, with a clear plateau in the progression-free survival curve [7,8]. However, despite these positive outcomes, HER2-positive metastatic breast cancer remains a biologically and clinically diverse disease [8]. Survival heterogeneity exists at a patient level, with some patients surviving only 1–2 years while others live for over a decade. 

Survival estimates from clinical trials without considering individual factors often fail to provide realistic prognoses. Large real-world studies examining survival outcomes specifically for de novo metastatic HER2-positive breast cancer are lacking, despite this subtype typically having a better prognosis than recurrent metastatic HER2-positive breast cancer. This study aims to fill that gap by providing real-world OS estimates and examining how demographic, clinicopathological, and treatment factors affect OS in patients with de novo metastatic HER2-positive breast cancer, using a large sample from the National Cancer Database (NCDB). 

## 2. Methods

This study utilized the NCDB, which includes data from over 1500 Commission on Cancer (CoC) programs that treat approximately 70% of all newly diagnosed cancers in the United States [3,4]. A retrospective analysis was conducted using de-identified patient records, following NCDB guidelines, qualifying the study for exemption from our institutional review board. The following variables were included in the analysis: demographics—age, race, and Charlson–Deyo comorbidity score (grouped as score 0, score 1, score 2, or score ≥3); clinicopathological variables—histology (invasive ductal, invasive lobular, favorable histology [adenoid cystic, cribriform, tubular, mucinous, and medullary], inflammatory, metaplastic, and other carcinomas), HER2 IHC expression, estrogen receptor (ER) status, progesterone receptor (PR) status, the number of metastatic sites (i.e., organs involved with metastatic cancer), and the location of metastases; and treatment variables—first-line chemotherapy, anti-HER2 therapy, hormone-blocking therapy, surgery of the primary site (breast), surgery of the non-primary sites, radiation, and palliative treatment (interventions to alleviate symptoms, such as surgery, radiation, systemic therapy, and/or pain management). A total of 5376 women diagnosed with de novo metastatic HER2-positive breast cancer between 2010 and 2020, with complete data available for all variables, were included in the analysis.

Demographic, clinicopathological, and treatment variables were summarized using frequencies and percentages. Overall survival (OS) was defined as the time from metastatic HER2-positive breast cancer diagnosis to death. Kaplan–Meier (KM) curves were used to estimate the 12-month, 36-month, and 60-month survival estimates (SE). The log-rank test evaluated OS differences between groups for each variable in the univariate analysis. Multivariate analysis used a Cox proportional hazard model with backward elimination to identify prognostic factors affecting OS. The 12-month, 36-month, and 60-month survival estimates, 95% confidence intervals (CIs), and adjusted hazard ratios (HR) were reported. PC SAS version 9.4 was used for all analyses. The statistical significance level was set at 0.05 for all analyses. 

## 3. Results

A total of 5376 women diagnosed with de novo metastatic HER2-positive breast cancer between 2010 and 2020 had a median OS of 55.95 months (95% CI 53.55-NE). Table 1 details the frequency of all demographic, clinicopathologic, and treatment characteristics. Most women were aged 41 to 70 years (67.2%), Caucasian (74%), and had a Charlson–Deyo comorbidity score of zero (82.4%). The majority had invasive ductal carcinoma (84.9%), had an HER2 IHC expression of 3+ (84.9%), were ER positive (61.6%), were PR negative (56%), had a single metastatic site (48.6%), and had extra-cranial metastases only (90.8%). Most women received single-agent first-line chemotherapy (46.5%), anti-HER2 therapy (78.8%), no hormone-blocking therapy (62.3%), no breast surgery (80.1%), no surgery of non-primary sites (94.6%), no radiation (68.4%), and no palliative treatment to alleviate symptoms (74.6%).

Table 2 presents the overall survival estimates for 12-month, 36-month, and 60-month periods and the hazard ratios and *p*-values for different groups based on demographic, clinicopathological, and treatment variables in the univariate analysis.

Demographic variables: The overall survival (OS) rates were lowest in the 71+ age group (HR 4.76, 95% CI 3.9–5.8) and were also lower in the 41–70 age group (HR 2.18, 95% CI 1.8–2.64) compared to the 19–40 age group. OS rates were lower in Blacks (HR 1.19, 95% CI 1.06–1.33) and higher in those classified as other (HR 0.79, 95% CI 0.66–0.95) than Caucasians. In contrast, OS rates were similar for those with an unknown race (HR 0.89, 95% CI 0.55–1.43) compared to Caucasians. Charlson–Deyo comorbidity score groups of one (HR 1.23, 95% CI 1.07–1.41), two (HR 2.36, 95% CI 1.93–2.88) and three or more (HR 3.72, 95% CI 3.11–4.45) were associated with worse OS rates compared to the score 0 group, demonstrating a clear trend of decreasing survival as the comorbidity burden increased.

Cancer-related clinicopathological variables: The overall survival (OS) rates were significantly worse for metaplastic carcinoma (HR 2.39, 95% CI 1.19–4.78) and other carcinoma (HR 1.99, 95% CI 1.75–2.27) compared to invasive ductal carcinoma (IDC). However, OS rates were similar for invasive lobular carcinoma (HR 1.11, 95% CI 0.89–1.37), favorable histology (HR 0.87, 95% CI 0.33–2.33), and inflammatory carcinoma (HR 1.17, 95% CI 0.82–1.68) when compared to invasive ductal carcinoma. Regarding HER2 IHC expression, women with 2 + and FISH+ (HR 1.22, 95% CI 1.09–1.37) had worse OS rates than those with 3+. ER-positive (HR 0.73, 95% CI 0.67–0.80) and PR-positive (HR 0.72, 95% CI 0.66–0.79) cancers were associated with better OS rates than ER-negative and PR-negative cancers. Regarding the number of metastatic sites, women with two (HR 1.44, 95% CI 1.30–1.61), three (HR 2.14, 95% CI 1.89–2.42), four (HR 2.52, 95% CI 2.15–2.96), and five or more (HR 4.02, 95% CI 3.25–4.97) metastatic sites had lower OS rates compared to those with only one metastatic site, showing a clear trend of decreasing survival as the number of metastatic sites increased. In a subgroup analysis among women with just a single metastatic site, those with lymph node-only (HR 0.30, 95% CI 0.20–0.46), bone-only (HR 0.37, 95% CI 0.25–0.53), liver-only (HR 0.42, 95% CI 0.29–0.62), and lung-only metastases (HR 0.45, 95% CI 0.30–0.67) had better OS rates than those with brain-only metastases, while OS rates were similar among those with other-only metastases (HR 0.84, 95% CI 0.53–1.32) and brain-only metastases. Women with extra-cranial-only metastases (HR 0.53, 95% CI 0.37–0.76) had better OS rates compared to those with cranial-only metastases, while OS rates were similar between cranial + extra-cranial metastases (HR 1.11, 95% CI 0.76–1.61) and cranial-only metastases.

Treatment characteristics: The OS rates were higher for those who received single-agent first-line chemotherapy (HR 0.25, 95% CI 0.22–0.27) or multi-agent first-line chemotherapy (HR 0.18, 95% CI 0.16–0.20) compared to no chemotherapy. This improvement in OS with first-line chemotherapy remained consistent across all Charlson–Deyo comorbidity score groups in the subgroup analysis; refer to Table 3 for details. Similarly, OS rates were higher for those who received anti-HER2 therapy (HR 0.22, 95% CI 0.20–0.24) than those who did not. This improvement in OS with anti-HER2 therapy remained consistent across all Charlson–Deyo comorbidity score groups in the subgroup analysis; refer to Table 4 for details. The OS rates were higher among those who received hormone-blocking therapy (HR 0.56, 95% CI 0.51–0.61) compared to those who did not. Women who underwent surgery for the primary tumor (breast) (HR 0.33, 95% CI 0.29–0.39) had higher OS rates than those who did not. Interestingly, surgery on a distant lymph node or metastatic site also improved OS rates (HR 0.64, 95% CI 0.50–0.82), while regional lymph node surgery had similar OS rates (HR 0.65, 95% CI 0.37–1.12) compared to no surgery. The OS rates were higher for those who received radiation therapy to the primary site (breast) (HR 0.54, 91% CI 0.43–0.68), radiation of local lymph nodes (HR 0.29, 95% CI 0.12–0.69), and radiation of distant lymph nodes or sites (HR 0.83, 95% CI 0.75–0.2) compared to no radiation at any site. However, receiving palliative treatment to alleviate symptoms was associated with lower OS rates (HR 1.29, 95% CI 1.18–1.42) compared to no palliative treatment.

In the multivariate Cox proportional hazard analysis, age, Charlson–Deyo comorbidity score, histology, HER2 IHC expression, ER expression, PR expression, the number of metastatic sites, metastasis location (intra-cranial only/intra-cranial and extra-cranial/extra-cranial only), and the receipt of first-line chemotherapy, anti-HER2 therapy, hormone-blocking therapy, surgery to the primary site (breast), surgery to the non-primary site, and palliative treatment were identified as independent factors influencing OS (Table 5). Race and radiation were not significant in the multivariate analysis. Figure 1 shows the Kaplan–Meier survival curves for the variables significant in the multivariate analysis. Figure 2 illustrates the prognostic factors and their impact on overall survival (OS). 

## 4. Discussion

Metastatic HER2-positive breast cancer prognostication from clinical trials may overestimate real-world outcomes due to stricter eligibility criteria, better health status, lower cancer burden, and greater treatment access among trial patients. Therefore, obtaining real-world overall survival (OS) estimates and understanding the factors that could influence OS is crucial [9,10]. Our study, the largest to date, included 5376 women with de novo metastatic HER2-positive breast cancer diagnosed between 2010 and 2020, revealing a real-world median OS of 55.95 months. Fortunately, this outcome is similar to the median OS observed in the first-line dual HER2-targeted therapy clinical trials [11]. It is important to note that our study focuses on de novo metastatic HER2-positive breast cancer only, which, in prior studies, has had better survival outcomes compared to recurrent or relapsed metastatic HER2-positive breast cancer, especially in patients with a metastasis-free interval of ≤24 months likely due to differences in demographics, cancer characteristics/ biology, and exposure to prior treatments [12,13].

In the multivariate analysis, we identified fourteen independent prognostic variables for survival in metastatic HER2-positive breast cancer: age, Charlson–Deyo score, histology, HER2 IHC expression, ER expression, PR expression, the number of metastatic sites, the location of metastases, first-line chemotherapy, anti-HER2 therapy, hormone-blocking therapy, surgery of the primary site (breast), surgery of the non-primary site, and palliative treatment (to alleviate symptoms). Race and radiation therapy were the only variables that were not significant in the multivariate analysis.

Our study revealed that age was an independent prognostic factor with an inverse correlation between age and OS. Prior studies of metastatic breast cancer (some of which included all types of metastatic breast cancer and not just HER2-positive cancers, as well as both de novo and recurrent metastatic cancer) have shown that median OS declines with increasing age at diagnosis, with one study indicating that for each additional year of age at diagnosis, the hazard of death increases by 3% [14,15,16]. Overall health significantly influences prognosis, as demonstrated by an inverse correlation between the Charlson–Deyo comorbidity score and OS in our study. A prior systematic review showed similar findings that in breast cancer and HER2-positive breast cancer, a score of ≥1 on the Charlson–Deyo comorbidity score is linked to lower OS compared to a score of zero. However, in that systematic review, among patients receiving systemic therapies (chemotherapy, hormonal therapy, or HER2-targeted therapy), there was no significant difference in OS for comorbidity scores of one or two versus zero, with significant declines in OS noted only for scores of three or higher [17]. This suggests that higher comorbidity scores can limit the ability to receive and tolerate systemic therapies, and as comorbidity scores increase, it may lead to decreased OS regardless of the treatment received.

Our study demonstrated that histology was also an independent prognostic factor, with similar OS rates in invasive ductal carcinoma, invasive lobular carcinoma, favorable histology, and inflammatory carcinoma, and worse OS rates in metaplastic carcinoma and other carcinomas. A prior study in metastatic HER2-positive breast cancer found differences in disease-specific survival and overall survival between invasive ductal carcinoma and invasive lobular carcinoma, depending on the site of metastatic involvement [5]. In our study, women with HER2 IHC expression of 3+ had better OS than those with 2+ and positive FISH. This finding was consistent with a prior study in localized HER2-positive breast cancer that showed survival outcomes were better in those with HER2 IHC 3+ than in those with 2+ and positive FISH; additionally, among those with HER2 IHC 2+ and positive FISH, patients with a HER2 copy number of ≥8 had better survival outcomes than those with low HER2 copy numbers [18]. Similar findings were observed in a prior study on metastatic HER2-positive breast cancer, where poor response to trastuzumab was linked to an HER2 IHC expression of 3+ in less than 75% of tumor cells or overall low-level or equivocal HER2 FISH amplification [19]. One possible explanation for the higher efficacy of HER2-targeted agents in patients with higher HER2 protein expression may be increased drug binding, enhanced antibody-dependent cellular cytotoxicity, or improved intracellular uptake of the HER2 agents. In our study, hormone receptor-positive status (ER-positive or PR-positive) emerged as an independent prognostic variable, indicating better overall survival (OS). Prior studies have also demonstrated that survival is better in HR+ and HER2-positive metastatic breast cancer compared to HR- and HER2-positive cases, likely due to distinct biological and growth patterns [16,20,21,22].

In our study, there was an inverse correlation between the number of metastatic sites and OS, likely due to the relationship between increased cancer burden and a higher number of metastatic sites. This finding was consistently observed in previous studies of metastatic breast cancer (all types) and in a study specifically examining HER2-positive metastatic breast cancer, showing that the number of metastatic sites has a prognostic influence on OS [16,23,24]. Prior studies of metastatic breast cancer have shown that individuals with brain metastases have poorer OS, a finding that was also observed in our study [16,24,25]. In our study, (1) among those with only a single metastatic site, individuals with brain (intra-cranial) metastases had worse OS than those with metastases in other sites; (2) regardless of the number of metastatic sites, individuals with only extra-cranial metastases had better OS compared to those with intra-cranial metastases alone or with both intra-cranial and extra-cranial metastases. The poor survival rates in individuals with brain metastases arise from several factors, including the brain’s unique microenvironment and the difficulties in effectively delivering systemic therapies across the blood-brain barrier. With newer HER2 agents, such as trastuzumab deruxtecan and tucatinib, which offer improved central nervous system penetration and effectiveness against brain metastases, there is hope that outcomes may improve for patients diagnosed after 2020.

In our study, the receipt of first-line chemotherapy, whether single-agent or multi-agent, which is a key component of the initial induction treatment for metastatic HER2-positive breast cancer, and the receipt of anti-HER2 therapy were identified as independent prognostic factors for overall survival (OS). Patients not receiving these therapies experienced worse OS, likely due to more rapid cancer progression. HER2-directed therapies have had a significant positive impact on OS in patients with HER2+ metastatic breast cancer, reshaping the prognostic relevance of HER2 to reflect a more favorable influence on survival in the era of HER2-targeted therapy [26]. Similarly, in our study, receiving hormone-blocking therapy was also an independent prognostic factor, highlighting the importance of comprehensively targeting both HR and HER2 receptors to achieve better outcomes, as demonstrated in prior studies [21]. The subgroup analysis of first-line chemotherapy and anti-HER2 therapy across all Charlson–Deyo score groups indicated that while more patients with higher Charlson–Deyo scores did not receive these therapies, those who did had better OS in all groups. This suggests that higher comorbidity scores should not automatically prevent the administration of chemotherapy or anti-HER2 therapy unless contraindications or serious toxicity concerns exist.

Currently, there is no strong evidence supporting surgery for metastatic breast cancer, as randomized studies (conducted on all types of metastatic breast cancer, not just HER2-positive metastatic breast cancer) have failed to demonstrate an overall survival benefit from surgery on the primary breast tumor [27]. Our study indicated that surgery on the primary breast site and distant site (distant lymph nodes or metastatic site) are independent prognostic factors with better OS in those who underwent surgery. With improved survival rates from systemic therapies in HER2-positive breast cancer, it is crucial to examine the usefulness of surgical interventions in HER2-positive metastatic breast cancer, particularly for oligometastatic patients. The potential advantages of these interventions cannot be overlooked without further studies. Surprisingly, our study revealed that the receipt of palliative treatments was associated with worse overall survival (OS), suggesting that while palliative treatments may alleviate symptoms and improve quality of life, they do not enhance OS. 

Although the univariate analysis revealed differences in overall survival (OS) based on race and the receipt of radiation therapy, these variables were not significant in the multivariate analysis. A previous study found that Black individuals had a slightly worse adjusted overall survival than White individuals (hazard ratio 1.29, 95% CI 1.00–1.65), which was not observed in our study [28]. Currently, there are no randomized studies demonstrating the survival benefit of radiation therapy in metastatic breast cancer; likewise, our study found that radiation therapy receipt was not a prognostic variable affecting OS in multivariate analysis [29].

One limitation of our study is that it enrolled patients diagnosed between 2010 and 2020. The Cleopatra trial results led to the FDA approval of pertuzumab in combination with docetaxel and trastuzumab in 2012, which remains the current standard for first-line treatment in metastatic breast cancer [30,31]. In subsequent years, the FDA approved other agents for HER2-positive metastatic breast cancer: trastuzumab emtansine in 2013 based on the EMILIA trial, Trastuzumab deruxtecan in 2019 based on the DESTINY-Breast01 trial, tucatinib in combination with trastuzumab and capecitabine in 2020 based on the HER2CLIMB trial, Margetuximab in combination with chemotherapy in 2020 based on the SOPHIA trial, and Neratinib in combination with capecitabine in 2020 based on the NALA trial [32,33,34,35,36]. These new anti-HER2 agents have significantly improved OS for metastatic HER2-positive breast cancer. OS outcomes for those diagnosed with metastatic HER2-positive breast cancer in 2012 or later are likely better; however, this is not fully reflected in our study’s survival estimates, which enrolled patients diagnosed from 2010 to 2020. Our results can serve as a benchmark to gauge progress in OS for de novo metastatic HER2-positive breast cancer in the coming decades. Additional limitations of our study arise from the NCDB, a hospital-based data collection system that aggregates information from various centers. This aggregation could lead to inconsistencies in data collection standards across different facilities, as well as potential errors during data abstraction and entry from medical records. Furthermore, the NCDB does not provide details on responses to first-line therapy, longitudinal treatment data (offering only information on first-line treatments without subsequent therapies), patient-reported outcomes such as quality of life, and information regarding causes of death.

## 5. Conclusions

In conclusion, this study is the largest and most comprehensive analysis of overall survival estimates in de novo metastatic HER2-positive breast cancer to date in a real-world setting. By evaluating a broad range of demographic, clinical, and treatment-related factors, we have identified several key independent prognostic variables that affect survival. Our findings will enable clinicians to offer personalized prognostic insights at diagnosis and customize treatment options. Additionally, this research lays the groundwork for establishing prognostic groups, potentially guiding future clinical trials in metastatic HER2-positive breast cancer.

## Figures and Tables

**Figure 1 cancers-17-01823-f001:**
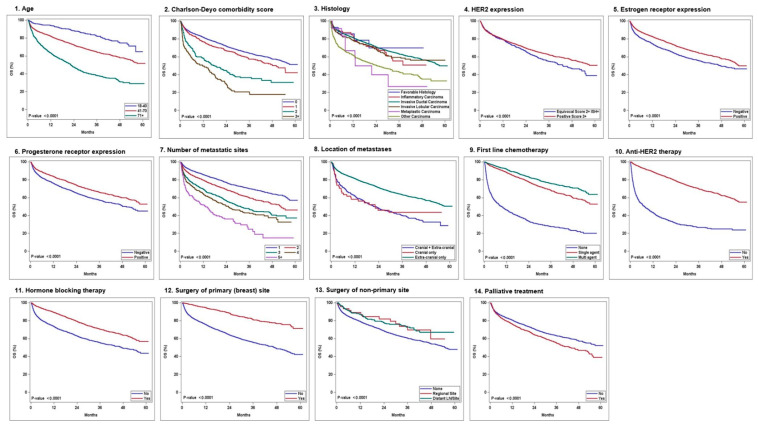
Kaplan–Meier survival curves of the significant variables in the multivariate analysis: 1—age, 2—Charlson—Deyo comorbidity score, 3—histology, 4—HER2 expression, 5—estrogen receptor expression, 6—progesterone receptor expression, 7—number of metastatic sites, 8—location of metastases, 9—first-line chemotherapy, 10—anti HER2 therapy, 11—hormone-blocking therapy, 12—surgery of primary (breast) site, 13—surgery of non-primary sites, and 14—palliative treatment.

**Figure 2 cancers-17-01823-f002:**
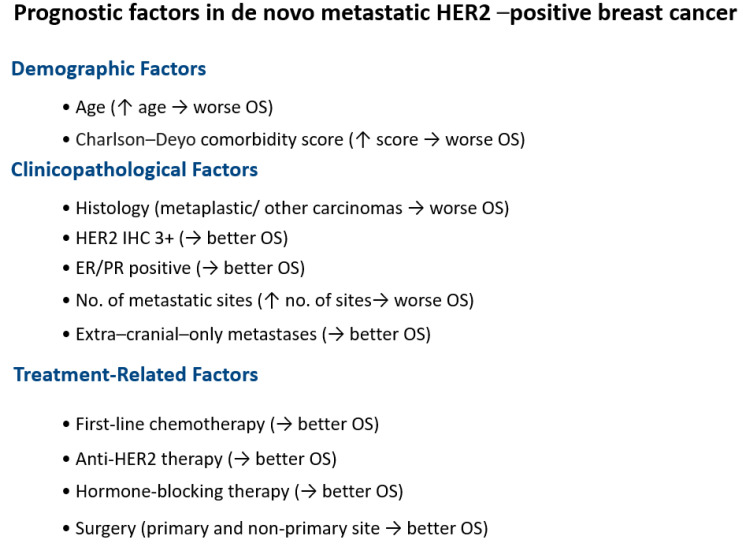
Schematic of prognostic factors in de novo metastatic HER2-positive breast cancer and their impact on OS. ↑ stands for increase. → stands for associated with.

**Table 1 cancers-17-01823-t001:** Demographic, clinicopathological, and treatment characteristics.

	No. of Patients, n (%)	No. of Events/n
Demographic characteristics		
Age		
19–40	633 (11.8%)	115/633
41–70	3611 (67.2%)	1239/3611
71+	1132 (21.1%)	651/1132
Race		
Caucasian	3976 (74%)	1472/3976
Black	951 (17.7%)	396/951
Other	395 (7.3%)	120/395
Unknown	54 (1%)	17/54
Charlson–Deyo score		
0	4432 (82.4%)	1533/4432
1	594 (11%)	238/594
2	170 (3.2%)	103/170
3 or more	180 (3.3%)	131/180
Clinicopathological/cancer characteristics		
Histology		
Invasive ductal carcinoma	4565 (84.9%)	1618/4565
Invasive lobular carcinoma	242 (4.5%)	88/242
Favorable histology	14 (0.3%)	4/14
Other carcinoma	465 (8.6%)	257/465
Inflammatory carcinoma	78 (1.5%)	30/78
Metaplastic carcinoma	12 (0.2%)	8/12
HER2 IHC expression		
2+ with FISH+	814 (15.1%)	351/814
3+	4562 (84.9%)	1654/4562
Estrogen receptor		
Negative	2063 (38.4%)	871/2063
Positive	3313 (61.6%)	1134/3313
Progesterone receptor		
Negative	3011 (56%)	1232/3011
Positive	2365 (44%)	773/2365
No. of metastatic sites		
1	2615 (48.6%)	754/2615
2	1492 (27.8%)	578/1492
3	779 (14.5%)	386/779
4	351 (6.5%)	191/351
5+	139 (2.6%)	96/139
Location of metastases		
Intra-cranial only	58 (1.1%)	31/58
Extra-cranial only	4880 (90.8%)	1717/4880
Intra-cranial + extra-cranial	438 (8.1%)	257/438
Treatment characteristics		
First-line chemotherapy		
None	1290 (24%)	869/1290
Single-agent	2502 (46.5%)	765/2502
Multi-agent	1584 (29.5%)	371/1584
Anti-HER2 therapy		
No	1141 (21.2%)	769/1141
Yes	4235 (78.8%)	1236/4235
Hormone-blocking therapy		
No	3351 (62.3%)	1419/3351
Yes	2025 (37.7%)	586/2025
Surgery of primary site (breast)		
No	4305 (80.1%)	1806/4305
Yes	1071 (19.9%)	199/1071
Surgery of non-primary site		
No	5087 (94.6%)	1927/5087
Regional nodes	46 (0.9%)	13/46
Distant lymph nodes orsites	243 (4.5%)54 (1%)	65/243
Radiation		
No	3675 (68.4%)	1437/3675
Primary site (breast)	292 (5.4%)	75/292
Local lymph nodes	34 (0.6%)	5/34
Distant lymph nodes or site	1375 (25.6%)	488/1375
Palliative treatment (to alleviate symptoms)		
No	4010 (74.6%)	1414/4010
Yes	1366 (25.4%)	591/1366

**Table 2 cancers-17-01823-t002:** Univariate analysis of demographic, clinicopathological, and treatment characteristics on OS.

	12-Month Survival Estimate (95% CI)	36-Month Survival Estimate (95% CI)	60-Month Survival Estimate (95% CI)	Hazard Ratio (95% CI)	Log-Rank *p*-Value
Demographic Characteristics					
Age					<0.0001
19–40	0.94 (0.92–0.96)	0.82 (0.79–0.86)	0.65 (0.53–0.80)	Reference	
41–70	0.82 (0.81–0.83)	0.64 (0.63–0.66)	0.52 (0.48–0.55)	2.18 (1.80–2.64)	
71+	0.63 (0.6–0.66)	0.38 (0.35–0.41)	0.29 (0.25–0.34)	4.76 (3.90–5.80)	
Race	0.8 (0.78–0.81)	0.61 (0.60–0.63)	0.5 (0.47–0.53)	Reference	0.0004
Caucasian	0.77 (0.74–0.8)	0.57 (0.54–0.61)	0.41 (0.34–0.5)	1.19 (1.06–1.33)	
Black					
Other	0.85 (0.82–0.89)	0.67 (0.62–0.72)	0.48 (0.35–0.67)	0.79 (0.66–0.95)	
Unknown	0.81 (0.71–0.92)	0.62 (0.48–0.79)	0.62 (0.48–0.79)	0.89 (0.55–1.43)	
Charlson					<0.0001
–Deyo score					
0	0.82 (0.81–0.83)	0.64 (0.63–0.66)	0.51 (0.48–0.55)	Reference	
1	0.77 (0.74–0.80)	0.57 (0.53–0.62)	0.42 (0.34–0.52)	1.23 (1.07–1.41)	
2	0.59 (0.51–0.67)	0.37 (0.30–0.45)	NE (NE–NE)	2.36 (1.93–2.88)	
3 or more	0.50 (0.43–0.58)	0.21 (0.15–0.29)	NE (NE–NE)	3.72 (3.11–4.45)	
Clinicopathologic/Cancer Characteristics					
Histology					<0.0001
Invasive ductal carcinoma	0.81 (0.80–0.82)	0.63 (0.62–0.65)	0.50 (0.47–0.53)	Reference	
Invasive lobular carcinoma	0.79 (0.74–0.84)	0.59 (0.53–0.67)	0.56 (0.49–0.64)	1.11 (0.89–1.37)	
Favorable histology	0.86 (0.69–1.00)	0.70 (0.49–1.00)	NE (NE–NE)	0.87 (0.33–2.33)	
Other carcinoma	0.62 (0.58–0.67)	0.43 (0.38–0.48)	0.33 (0.27–0.40)	1.99 (1.75–2.27)	
Inflammatory carcinoma	0.83 (0.75–0.92)	0.55 (0.44–0.70)	NE (NE–NE)	1.17 (0.82–1.68)	
Metaplastic carcinoma	0.67 (0.45–0.99)	0.27 (0.09–0.78)	NE (NE–NE)	2.39 (1.19–4.78)	
HER2 IHC expression					0.0007
3+	0.80 (0.78–0.81)	0.62 (0.61–0.64)	0.50 (0.47–0.54)	Reference	
2+ with FISH+	0.79 (0.77–0.82)	0.55 (0.51–0.59)	0.39 (0.32–0.47)	1.22 (1.09–1.37)	
Estrogen receptor					<0.0001
Negative	0.74 (0.73–0.76)	0.56 (0.54–0.58)	0.46 (0.43–0.50)	Reference	
Positive	0.83 (0.81–0.84)	0.64 (0.63–0.66)	0.50 (0.46–0.54)	0.73 (0.67–0.80)	
Progesterone receptor					<0.0001
Negative	0.76 (0.75–0.78)	0.57 (0.55–0.59)	0.45 (0.42–0.49)	Reference	
Positive	0.84 (0.82–0.85)	0.66 (0.64–0.68)	0.53 (0.48–0.58)	0.72 (0.66–0.79)	
No. of metastatic sites					<0.0001
1	0.86 (0.84–0.87)	0.70 (0.68–0.72)	0.57 (0.52–0.62)	Reference	
2	0.80 (0.78–0.82)	0.60 (0.57–0.63)	0.46 (0.41–0.52)	1.44 (1.30–1.61)	
3	0.70 (0.66–0.73)	0.47 (0.43–0.51)	0.37 (0.31–0.44)	2.14 (1.89–2.42)	
4	0.65 (0.60–0.70)	0.43 (0.38–0.49)	NE (NE–NE)	2.52 (2.15–2.96)	
5+	0.52 (0.44–0.61)	0.25 (0.17–0.36)	NE (NE–NE)	4.02 (3.25–4.97)	
Site of metastatic involvement (subgroup analysis in those with single metastatic site)					<0.0001
Brain only	0.58 (0.47–0.72)	0.44 (0.32–0.59)	NE (NE–NE)	Reference	
LN only	0.94 (0.91–0.96)	0.76 (0.71–0.81)	0.61 (0.46–0.80)	0.30 (0.20–0.46)	
Bone only	0.88 (0.86–0.90)	0.73 (0.70–0.76)	0.57 (0.50–0.65)	0.37 (0.25–0.53)	
Liver only	0.83 (0.80–0.86)	0.71 (0.67–0.75)	0.56 (0.48–0.66)	0.42 (0.29–0.62)	
Lung only	0.85 (0.82–0.89)	0.66 (0.61–0.72)	0.61 (0.55–0.67)	0.45 (0.30–0.67)	
Other only	0.71 (0.63–0.81)	0.48 (0.38–0.6)	0.40 (0.29–0.57)	0.84 (0.53–1.32)	
Location of metastases					<0.0001
Intra-cranial only	0.58 (0.47–0.72)	0.44 (0.32–0.59)	NE (NE–NE)	Reference	
Extra-cranial only	0.81 (0.80–0.82)	0.63 (0.62–0.65)	0.50 (0.47–0.54)	0.53 (0.37–0.76)	
Intra-cranial + extra-cranial	0.62 (0.58–0.67)	0.40 (0.36–0.45)	NE (NE–NE)	1.11 (0.76–1.61)	
Treatment characteristics					
First-line chemotherapy					<0.0001
None	0.46 (0.44–0.49)	0.28 (0.25–0.30)	0.20 (0.16–0.25)	Reference	
Single-agent	0.88 (0.87–0.90)	0.68 (0.66–0.70)	0.53 (0.48–0.58)	0.25 (0.22–0.27)	
Multi-agent	0.92 (0.91–0.93)	0.76 (0.74–0.78)	0.64 (0.58–0.69)	0.18 (0.16–0.20)	
Anti-HER2 therapy					<0.0001
No	0.43 (0.41–0.46)	0.27 (0.24–0.30)	0.24 (0.20–0.28)	Reference	
Yes	0.89 (0.88–0.90)	0.70 (0.68–0.71)	0.55 (0.52–0.59)	0.22 (0.20–0.24)	
Hormone-blocking therapy					<0.0001
No	0.74 (0.72–0.75)	0.55 (0.54–0.57)	0.44 (0.40–0.48)	Reference	
Yes	0.89 (0.88–0.91)	0.70 (0.68–0.73)	0.57 (0.52–0.61)	0.56 (0.51–0.61)	
Surgery of primary site (breast)					<0.0001
No	0.76 (0.74–0.77)	0.56 (0.54–0.58)	0.42 (0.39–0.46)	Reference	
Yes	0.95 (0.93–0.96)	0.81 (0.78–0.84)	0.71 (0.66–0.77)	0.33 (0.29–0.39)	
Surgery of non-primary site					0.0005
No	0.79 (0.78–0.80)	0.60 (0.59–0.62)	0.48 (0.45–0.51)	Reference	
Regional nodes	0.89 (0.81–0.99)	0.70 (0.56–0.86)	NE (NE–NE)	0.65 (0.37–1.12)	
Distant lymph nodes or site	0.87 (0.83–0.91)	0.73 (0.67–0.79)	NE (NE–NE)	0.64 (0.50–0.82)	
Radiation					<0.0001
No	0.77 (0.76–0.79)	0.59 (0.57–0.61)	0.45 (0.42–0.49)	Reference	
Primary site (breast)	0.94 (0.91–0.97)	0.72 (0.66–0.78)	0.65 (0.58–0.74)	0.54 (0.43–0.68)	
Local lymph nodes	0.97 (0.92–1.00)	0.85 (0.74–0.98)	NE (NE–NE)	0.29 (0.12–0.69)	
Distant lymph nodes or site	0.82 (0.80–0.84)	0.64 (0.61–0.66)	0.52 (0.47–0.58)	0.83 (0.75–0.92)	
Palliative treatment (to alleviate symptoms)					<0.0001
No	0.81 (0.79–0.82)	0.63 (0.61–0.65)	0.52 (0.49–0.56)	Reference	
Yes	0.76 (0.74–0.78)	0.55 (0.53–0.58)	0.39 (0.34–0.45)	1.29 (1.18–1.42)	

**Table 3 cancers-17-01823-t003:** Sub-group analysis of first-line chemotherapy receipt in each Charlson–Deyo comorbidity score group.

**First-Line Chemotherapy (No. of Patients, n)**	**12-Month Survival Estimate** **(95% CI)**	**36-Month Survival Estimate** **(95% CI)**	**60-Month Survival Estimate** **(95% CI)**	**Hazard Ratio** **(95% CI)**	**Log-Rank *p*-Value**
Charlson–Deyo comorbidity score 0; No. of patients = 4432 (82.4%)					
None (n = 961)	0.49 (0.46–0.52)	0.30 (0.27–0.34)	0.22 (0.18–0.27)	Reference	<0.0001
Single-agent (n = 2123)	0.89 (0.88–0.91)	0.70 (0.68–0.72)	0.54 (0.49–0.60)	0.25 (0.23–0.28)	
Multi-agent (n = 1348)	0.93 (0.91–0.94)	0.78 (0.75–0.80)	0.66 (0.61–0.72)	0.18 (0.15–0.20)	
Charlson–Deyo comorbidity score 1; No. of patients = 594 (11%)					
None (n = 157)	0.45 (0.38–0.54)	0.28 (0.21–0.37)	NE (NE–NE)	Reference	<0.0001
Single-agent (n = 253)	0.85 (0.81–0.90)	0.65 (0.59–0.72)	0.49 (0.38–0.64)	0.26 (0.20–0.35)	
Multi-agent (n = 184)	0.92 (0.88–0.96)	0.72 (0.65–0.79)	0.48 (0.33–0.71)	0.21 (0.15–0.29)	
Charlson–Deyo comorbidity score 2; No. of patients = 170 (3.2%)					
None (n = 76)	0.32 (0.23–0.45)	0.14 (0.08–0.26)	NE (NE–NE)	Reference	<0.0001
Single-agent (n = 63)	0.81 (0.71–0.91)	0.58 (0.46–0.73)	NE (NE–NE)	0.25 (0.16–0.40)	
Multi-agent (n = 31)	0.77 (0.64–0.94)	0.48 (0.32–0.71)	NE (NE–NE)	0.29 (0.17–0.52)	
Charlson–Deyo comorbidity score = 3 or more; No. of patients = 180 (3.3%)					
None (n = 96)	0.32 (0.24–0.43)	0.10 (0.05–0.20)	NE (NE–NE)	Reference	<0.0001
Single-agent (n = 63)	0.72 (0.62–0.84)	0.29 (0.9–0.45)	NE (NE–NE)	0.44 (0.30–0.64)	
Multi-agent (n = 21)	0.61 (0.43–0.86)	0.49 (0.31–0.78)	NE (NE–NE)	0.34 (0.18–0.66)	

**Table 4 cancers-17-01823-t004:** Sub-group analysis of anti-HER2 therapy receipt in each Charlson–Deyo comorbidity score group.

**Anti-HER2 Therapy (No. of Patients, n)**	**12-Month Survival Estimate** **(95% CI)**	**36-Month Survival Estimate** **(95% CI)**	**60-Month Survival Estimate** **(95% CI)**	**Hazard Ratio** **(95% CI)**	**Log-Rank *p*-Value**
Charlson–Deyo comorbidity score 0; No. of patients = 4432 (82.4%)					
No (n = 859)	0.47 (0.44–0.51)	0.30 (0.27–0.34)	0.27 (0.23–0.31)	Reference	<0.0001
Yes (n = 3573)	0.90 (0.89–0.91)	0.72 (0.70–0.74)	0.57 (0.53–0.61)	0.22 (0.20–0.25)	
Charlson–Deyo comorbidity score 1; No. of patients = 594 (11%)					
No (n = 139)	0.41 (0.33–0.50)	0.24 (0.17–0.34)	NE (NE–NE)	Reference	<0.0001
Yes (n = 455)	0.88 (0.85–0.91)	0.67 (0.63–0.72)	0.48 (0.39–0.60)	0.22 (0.17–0.29)	
Charlson–Deyo comorbidity score 2; No. of patients= 170 (3.2%)					
No (n = 58)	0.26 (0.16–0.40)	0.11 (0.04–0.26)	NE (NE–NE)	Reference	<0.0001
Yes (n = 112)	0.75 (0.68–0.84)	0.50 (0.41–0.61)	NE (NE–NE)	0.23 (0.15–0.34)	
Charlson–Deyo comorbidity score = 3 or more; No. of patients = 180 (3.3%)					
No (n = 85)	0.23 (0.15–0.34)	0.10 (0.05–0.20)	NE (NE–NE)	Reference	<0.0001
Yes (n = 95)	0.73 (0.64–0.83)	0.30 (0.21–0.43)	NE (NE–NE)	0.31 (0.21–0.44)	

**Table 5 cancers-17-01823-t005:** Multivariate Cox proportional hazard analysis.

	Hazard Ratio (95% CI)	Overall *p*-Value
Age		<0.0001
19–40	Reference
41–70	1.56 (1.28–1.89)
71+	2.19 (1.78–2.71)
Charlson–Deyo score		<0.0001

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

## Data Availability

The original contributions presented in this study are included in the article. Further inquiries can be directed to the corresponding author.

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
