# Peer review of "Overall Survival and Prognostic Factors in De Novo Metastatic Human Epidermal Growth Factor Receptor (HER)-2-Positive Breast Cancer: A National Cancer Database Analysis"

_cancers, 2025, doi:10.3390/cancers17111823_

Round 1
Reviewer 1 Report
Comments and Suggestions for Authors
This is a clinically relevant and well-powered study that utilizes NCDB data to explore overall survival and prognostic factors in de novo HER2-positive metastatic breast cancer (MBC). The findings are of interest to the oncology community, especially given the rising incidence of de novo MBC and expanding HER2-targeted treatment landscape. However, several scientific aspects require further clarification or deeper analysis.
- The observation that HER2 IHC 3+ confers better survival than 2+/FISH+ is interesting but inadequately explained. Beyond stating statistical significance, the authors may explore potential biological underpinnings—such as differences in HER2 protein density affecting antibody-dependent cellular cytotoxicity (ADCC) or intracellular trafficking affecting ADC uptake. Are there prior data suggesting that HER2-low tumors (2+/FISH+) respond suboptimally to trastuzumab-based therapy in the metastatic setting?
- While the paper stratifies survival by hormone receptor (HR) status, the explanation is superficial. HR+/HER2+ tumors are known to exhibit lower pathologic complete response (pCR) but longer survival due to endocrine sensitivity. In contrast, HR−/HER2+ (HER2-enriched or basal-like) tumors often show high proliferation and pCR but shorter relapse-free survival. Can the authors discuss whether these subtypes differ in de novo MBC biology and if this translates to differential response to HER2-targeted agents in the metastatic setting?
- The authors identify brain metastasis as an independent poor prognostic factor, but stop short of linking this to HER2 biology. Since HER2-positive tumors have a known predilection for CNS spread, the authors may briefly discuss whether newer agents with CNS penetration (e.g., tucatinib, trastuzumab deruxtecan) may alter the observed survival pattern in more contemporary datasets. Are brain mets associated with worse outcomes only due to site or due to treatment limitations?
- The authors note that systemic therapy was associated with better OS, but do not specify trastuzumab usage rates. Since NCDB lacks this detail, it would be useful to discuss how this limitation affects interpretation. Could part of the variation in outcome across HER2 IHC levels or HR subtypes reflect differential access to trastuzumab or HER2-targeted regimens?
- Does the prognostic impact of HER2 status differ by site of metastasis? For instance, is the survival benefit of HER2 IHC 3+ retained in patients with visceral vs non-visceral metastasis? A stratified analysis could uncover context-specific effects and improve clinical applicability.
- Though the focus is de novo cases, the discussion omits comparison with recurrent MBC. Literature suggests de novo MBC may have distinct biology and better response to HER2 blockade. Could the authors contrast their findings with recurrent HER2+ cohorts from other studies?
- In the Cox model, certain variables such as race and insurance show significance. Are these acting as surrogates for treatment access? The authors may carefully interpret whether these are independent prognostic factors or confounded by unmeasured variables such as delayed diagnosis or treatment non-compliance.
- Given the clinical and mechanistic relevance of the findings, a schematic illustration summarizing the core prognostic model—including HER2 IHC levels, HR status, metastasis sites, and treatment impact, that would be highly valuable. Readers will benefit from a figure that conveys this framework visually. Such practice is common in high-impact articles and enhances clarity, especially in real-world datasets.
https://www.mdpi.com/2227-7390/9/21/2814
https://www.mdpi.com/2077-0383/13/11/3353
https://www.mdpi.com/2072-6694/17/8/1261
Reviewer 2 Report
Comments and Suggestions for Authors
The manuscript entitled “Overall Survival and Prognostic Factors in De-novo Metastatic Human Epidermal Growth Factor Receptor (HER)-2-Positive Breast Cancer: A National Cancer Database Analysis” presents a well-executed real-world study analyzing the overall survival and key prognostic factors associated with de novo metastatic HER2-positive breast cancer. The study utilizes a large dataset and identifies relevant clinical parameters influencing survival, such as patient age, histology, hormone receptor status, and treatment modalities. The manuscript is well-written and offers valuable insights. I would recommend the acceptance of this paper if the authors could address my following comments.
1. Please expand the introduction section by including a brief overview of breast cancer, including its molecular subtypes, current treatment regimens, and challenges such as treatment resistance and disease progression.
2. It would be better if the authors could add a flowchart diagram or other type of diagram to detailly illustrate the methods, including the study design, case selection process, and inclusion/exclusion criteria. A visual depiction will provide readers with a clearer understanding of the analytical process.
3. For the results section, it would be better if the authors could reorganize and separate this section into clear subsections to improve readability.
4. Please expand the Figure 1 legend by detailing what each sub-panel represents.
Author Response
We thank the reviewer for their valuable time and comments on improving the quality of our paper.
Comment 1. Please expand the introduction section by including a brief overview of breast cancer, including its molecular subtypes, current treatment regimens, and challenges such as treatment resistance and disease progression.
Reply 1: This has been added to the introduction section- highlighted in red.
Comment 2. It would be better if the authors could add a flowchart diagram or other type of diagram to detailly illustrate the methods, including the study design, case selection process, and inclusion/exclusion criteria. A visual depiction will provide readers with a clearer understanding of the analytical process.
Reply 2: Thank you for the comment. The authors do not feel that adding a flow chart diagram to illustrate the methods adds any meaningful value, as it is a retrospective analysis using the NCDB database. We have selected patients with metastatic HER2-positive breast cancer from the NCDB database who had information regarding all the variables being analyzed in the study. This has been mentioned in the methods section, and we do not have any further information to add/ depict in the flow diagram.
Comment 3. For the results section, it would be better if the authors could reorganize and separate this section into clear subsections to improve readability.
Reply 3: This has been organized into subsections, as suggested. The first paragraph is a general description of the results and the most common distribution in each variable. Univariate analysis: The second paragraph concerns the OS in the various demographic variables, the third paragraph concerns clinicopathological variables, and the fourth paragraph concerns treatment variables. The fifth paragraph concerns results from the multivariate analysis.
Comment 4. Please expand the Figure 1 legend by detailing what each sub-panel represents.
Reply 4: This has been ade thrid paragraph: ded.
Reviewer 3 Report
Comments and Suggestions for Authors
In this study, Kesireddy et al. performed an overall survival analysis of a de novo metastatic HER2-positive breast cancer dataset obtained from the National Cancer Database (NCDB). The authors identified several prognostic factors contributing to the overall survival of the study population. This work is highly significant and makes a valuable contribution to the current understanding of HER2-positive breast cancer prognosis. The authors have conducted a thorough analysis using appropriate statistical tests. I do not have any specific comments or suggestions and recommend the manuscript for publication.
Author Response
Comment 1: In this study, Kesireddy et al. performed an overall survival analysis of a de novo metastatic HER2-positive breast cancer dataset obtained from the National Cancer Database (NCDB). The authors identified several prognostic factors contributing to the overall survival of the study population. This work is highly significant and makes a valuable contribution to the current understanding of HER2-positive breast cancer prognosis. The authors have conducted a thorough analysis using appropriate statistical tests. I do not have any specific comments or suggestions and recommend the manuscript for publication.
Reply 1: We thank the reviewer for their valuable time and positive feedback.